# Precautionary Health Behaviours as Potential Confounders in COVID-19 Vaccine Effectiveness Studies

**DOI:** 10.3390/vaccines13101047

**Published:** 2025-10-12

**Authors:** Chloé Wyndham-Thomas, Leonie de Munter, Kok Yew Ngew, Sanskruti Gaikwad, Konstantina Chatzikonstantinidou, Antonio Carmona, Charlotte Martin, Gerrit Luit ten Kate, Nicolas Praet, Wilhelmine Meeraus, Matthew D. Rousculp, Kaatje Bollaerts

**Affiliations:** 1P95 Clinical and Epidemiology Services, Diestsevest 125, 3000 Leuven, Belgium; chloe.wyndhamthomas@fagg-afmps.be (C.W.-T.); kok.yewngew@p-95.com (K.Y.N.); konstantina.chatzikonstantinidou@p-95.com (K.C.); kaatje.bollaerts@p-95.com (K.B.); 2P95 Clinical and Epidemiology Services, Weena 690, 3012 CN Rotterdam, The Netherlands; sanskruti.gaikwad@p-95.com; 3Fundación Fomento de la Investigación Sanitaria y Biomédica de la Comunidad Valenciana, Av. de Catalunya 21, 46020 Valencia, Spain; toni.carmona@p-95.com; 4Centre Hospitalier Universitaire St Pierre, Rue Haute 322, 1000 Brussels, Belgium; charlotte.martin@stpierre-bru.be; 5Universitair Ziekenhuis Antwerpen, Drie Eikenstraat 655, 2650 Edegem, Belgium; gerritluit.tenkate@uza.be; 6Janssen Research & Development, LLC, 2340, Turnhoutseweg 30, 2340 Beerse, Belgium; npraet@its.jnj.com; 7Medical Evidence, Vaccines & Immune Therapies, AstraZeneca, Cambridge CB2 8DU, UK; wilhelmine.meeraus@astrazeneca.com; 8Novavax, Inc., Gaithersburg, MD 20878, USA; mrousculp@novavax.com

**Keywords:** vaccine effectiveness, adults, SARS-CoV-2, confounders, bias, Europe, COVID-19

## Abstract

**Background/Objectives**: Precautionary health behaviours (PHBs), such as hand-washing or self-isolation, are non-pharmaceutical interventions used to reduce SARS-CoV-2 transmission. We investigated the potential confounding by PHBs of COVID-19 vaccine effectiveness (VE) estimates in a subset of study participants enrolled in id.DRIVE. **Methods**: The id.DRIVE COVID-19 VE study (formerly COVIDRIVE) is a European multicentre test-negative case–control study estimating COVID-19 VE against hospitalisation due to laboratory-confirmed SARS-CoV-2 in patients with severe acute respiratory infection. All adults (≥18 y) prospectively enrolled between 16 November 2021 and 16 August 2023 at three sites were invited to complete a PHB survey capturing indicators of PHBs in the 3 months preceding admission. Fisher’s exact test with Bonferroni-adjusted threshold was used to measure the level of association between PHB indicators and both COVID-19 vaccine status and SARS-CoV-2 test result. VE estimates were generated with and without adjustment for PHBs. **Results**: PHBs were modified over time, with higher precautionary attitudes in the first COVID-19 vaccine booster season (2021–2022) compared to the second one (2022–2023). For the first booster season, PHBs were positively associated with exposures (vaccination status) and outcomes (case or control status). Adjusting for PHBs led to a 6 to 9 percentage-point increase in VE estimates. Conversely, no confounding by PHBs was observed in the second booster season. **Conclusions**: PHBs should be considered a possible confounder of COVID-19 VE studies. Further research is needed to define when PHBs should be integrated into VE models, as the level of confounding may differ according to the study population and the epidemiological context.

## 1. Introduction

The development of safe and effective vaccines against coronavirus disease 2019 (COVID-19) was a pillar in the severe acute respiratory syndrome coronavirus 2 (SARS-CoV-2) pandemic response, with an estimated 1.6 million lives saved in the World Health Organization European Region between December 2020 and March 2023 [1]. These COVID-19 vaccines remain an important preventive measure in the post-pandemic era and are now used for seasonal immunisation of individuals at the highest risk of severe SARS-CoV-2 infection. To continuously assess the performance of these vaccines in real-world conditions and fulfil post-authorisation regulatory obligations of vaccine manufacturers, vaccine effectiveness (VE) studies are performed, and their results guide both vaccine development and vaccine policies.

Beside vaccination, non-pharmaceutical interventions are other public health measures that have been used in the fight against COVID-19 [2]. They encompass a wide spectrum of government-, medically, or self-initiated activities that aim to interrupt infection chains. For COVID-19, these have varied from social distancing interventions (wide-scale population lockdowns, prohibition of gatherings, border closures, school closures, etc.) to precautionary health behaviours (PHBs) such as hand-washing, respiratory etiquette, wearing a face mask, and self-isolation [3].

PHBs may act as a confounder in COVID-19 VE studies. Individuals adhering most strongly to PHBs have shown higher vaccine willingness, suggesting a possible association between PHBs and exposure [4]. Because PHBs are preventive measures that reduce the risk of being exposed to and infected by SARS-CoV-2 [5], they are also likely associated with the outcome. However, more complex interactions may be involved as the drivers of health behaviours, such as disease knowledge, disease perception, and public health preventive policies, which have been modified over the course of the SARS-CoV-2 pandemic.

Overall, the level of confounding by PHBs of COVID-19 VE estimates is rarely studied and COVID-19 VE reference protocols do not account for PHBs [6]. Here, we present the results of a PHB survey carried out in a subset of participants enrolled in the id.DRIVE COVID-19 VE study. To assess for potential confounding, the level of association between PHBs captured in the survey and both the exposure and the outcome was measured for different study periods, and COVID-19 VE estimates were generated in a post hoc analysis with and without adjustment for PHBs.

## 2. Materials and Methods

### 2.1. Study Design

This study was based on a survey collecting PHB data from a subset of id.DRIVE COVID-19 VE study participants at three out of eight id.DRIVE recruiting sites. The id.DRIVE COVID-19 VE study (EUPAS422328; formerly named COVIDRIVE) is an ongoing multi-country, hospital-based study using a test-negative case–control (TNCC) design to monitor brand-specific COVID-19 VE against hospitalisation in Europe. The detailed methodology has been previously described [7], and the latest study protocol is available at https://iddrive.eu/.

### 2.2. Study Population and Setting

Three out of eight id.DRIVE recruiting sites (Study Contributors) had the necessary local capacity to participate in the PHB survey. These sites were the Italian network Centro Interuniversitario per la Ricerca sull’Influenza e le altre Infezioni Transmissibili (CIRI-IT, Italy; a network of three hospital sites located in Rome, Bari, and Genoa), Centre Hospitalier Universitaire St Pierre (CHU St Pierre, Brussels, Belgium), and Universitair Ziekenhuis Antwerpen (UZA, Antwerp, Belgium). Adults 18 years and older prospectively enrolled in the COVID-19 VE study between 16 November 2021 and 16 August 2023 were invited to complete a PHB survey. According to the id.DRIVE inclusion criteria, the study population comprised patients hospitalised for at least an overnight stay with severe acute respiratory infection (SARI). SARI was defined as a suspicion of respiratory infection and at least one of the following symptoms—cough, fever, shortness of breath, or sudden onset of anosmia, ageusia, or dysgeusia—with symptom onset occurring within the last 14 days before hospital admission (modified from the European Centre for Disease Prevention and Control [ECDC] SARI case definition) [6]. Patients hospitalised for COVID-19 in the 3 months before the current admission and patients vaccinated with a non-EMA-approved vaccine were excluded.

### 2.3. Precautionary Health Behaviour Survey

Patients could undertake the survey at any time during their hospital stay, independently of the timing of SARS-CoV-2 testing and results. Data on PHBs were collected through a self-assessment questionnaire modified from Iorfa et al. [8] (Appendix A). The questionnaire included seven statements relating to PHBs in the 3 months before admission. Each statement had a three-point Likert scale (strongly agree, undecided, strongly disagree). Five statements were indicative of PHBs (e.g., I would self-isolate myself at home if needed), and two statements were indicative of non-PHBs (e.g., I do not mind going to very crowded places).

An indicator of PHBs was defined as a statement meeting its highest score: strongly agree for statements indicative of a PHB and strongly disagree for statements indicative of a non-PHB. The sum of PHB indicators per completed questionnaire determined the survey participants’ PHB composite scores ranging from 0 (no statements were indicators of PHBs) to 7 (all statements were indicators of PHBs).

### 2.4. Study Periods and Definitions

To compare changes over time, two study periods were defined: the first COVID-19 vaccine booster season, from 16 November 2021 (start date of survey data collection) to 31 August 2022, and the second COVID-19 vaccine booster season, from 1 September 2022 to 16 August 2023 (end date of survey data collection).

#### 2.4.1. Exposure Definitions

Vaccinated in the study period was defined as a survey participant having received either a complete primary schedule or at least one booster dose with any COVID-19 vaccine during the study period of interest, and before the onset of symptoms that led to SARI hospitalisation. Recently vaccinated survey participants (i.e., the last COVID-19 dose was received <14 days before symptom onset) and those with an incomplete primary schedule (i.e., only the first dose was received for COVID-19 vaccines administered as a two-dose priming schedule) were excluded from analysis. Unvaccinated in the study period was defined as a survey participant having (1) no history of COVID-19 vaccine dose (i.e., never vaccinated) or (2) having received the last dose before the study period of interest and ≥6 months before symptom onset (i.e., not vaccinated during the latest vaccine campaign and protection of any previous vaccinations has likely waned [9]).

#### 2.4.2. Outcome Definitions

Test-positive cases were survey participants meeting the SARI case definition and testing positive for SARS-CoV-2 on any respiratory specimen collected within 14 days before and up to 72 h after hospital admission. Test-negative controls were survey participants meeting the SARI case definition and negative for SARS-CoV-2 at hospital admission and on all tests collected in the 14 days preceding admission and all tests collected up to 72 h after hospital admission. RT-PCR or other RNA amplification systems with equivalent sensitivity were used for SARS-CoV-2 testing.

### 2.5. Statistical Methods

Only survey participants with fully completed questionnaires were included in these analyses. Owing to the inherent nature of this post hoc analysis, no specific sample size justification was provided to ensure sufficient statistical power. Instead, the analysis included all patients enrolled in the study who met the criteria outlined above. The analysis reported observations as they were, without imputing missing data. In analyses involving modelling, a complete-case analysis approach was used.

A variable is considered a confounder if the following three criteria are met: (1) the variable must be associated with the exposure; (2) the variable must cause the outcome; and (3) the variable cannot be on the causal pathway between exposure and outcome.

Associations between indicators of PHBs and exposure and outcome were measured using Fisher’s exact test. Bonferroni adjustment was applied, with the significance level of the *p*-value set at 0.0036 (0.05 divided by 14 statistical tests: 7 tests for exposure and 7 for outcome).

VE estimates: COVID-19 VE against hospitalisation was estimated in SARI patients as VE = (1 − OR) × 100% [10], where the odds ratio (OR) compared the odds of being vaccinated among the test-positive cases to the odds among the test-negative controls. Generalised estimating equation (GEE) models were developed to compare VE estimates with and without adjustment for PHBs. The reference model, following id.DRIVE COVID-19 VE study methods, was adjusted for symptom-onset date, sex, age, and number of chronic conditions among a predefined list of comorbidities known to increase the risk of severe COVID-19 (asthma, lung disease, cardiovascular disease, hypertension, chronic kidney disease, chronic liver disease, type 2 diabetes, cancer, and immunodeficiency). The comparative model was further adjusted for PHBs, fitting the PHB composite score either as (1) a continuous variable (PHB-adjusted model 1) or (2) a binary covariate using a cut-off value of ≥5, based on the median of the PHB composite score (PHB-adjusted model 2). VE was reported with a 95% confidence interval (CI). The changes in VE between the reference model and the PHB-adjusted models were summarised as percentages. The goodness-of-fit of the reference model and the PHB-adjusted models were compared using the likelihood ratio test, with a *p*-value of <0.05 as the threshold for significance. As a sensitivity analysis, VE estimates were calculated using generalised additive models (GAMs) instead of GEEs, with Study Contributor included as an additional covariate. When using GAMs, the goodness-of-fit values of the reference model and the PHB-adjusted model were compared using the likelihood ratio test. In addition, GEE-based VE estimates using the PHB-adjusted model 2 were generated using cut-offs of ≥4 and ≥6 rather than ≥5.

### 2.6. Ethics

The id.DRIVE study was conducted in accordance with the Declaration of Helsinki. The id.DRIVE COVID-19 VE study (EUPAS422328) protocol, which includes the PHB survey, was submitted and approved by the local independent ethics committees of each study site. At Belgian sites, the Central Ethics Committee approval was valid for both CHU Saint Pierre and UZA (B3002021000252, 10 January 2022). At Italian sites, the following Ethics Committees approved the conduct of the study: Comitato Etico Regionale della Liguria, Genoa (11528, 4 May 2021), Cominato Etico Indipendente Azienda Ospedaliero-Universitaria “Consorziale Policlinico”, Bari (6960, 4 August 2021), and Comitato Etico dell’Università “Sapienza”, Rome (6490_2021, 16 September 2021). Informed consent was provided by all participants or their legally acceptable representative(s). This post hoc analysis was approved by the id.DRIVE Steering Committee.

## 3. Results

### 3.1. Enrolment and Survey Participation

Between 16 November 2021 and 16 August 2023, three sites prospectively enrolled 1638 SARI patients in the id.DRIVE COVID-19 VE study. Of them, 1433 patients (87%) participated in the PHB survey, with a complete questionnaire obtained from 1427. Survey participation rates varied widely between the sites, from 38% (70/182) at UZA to 95% (1262/1332) at CIRI-IT, and were higher during the first booster season (90%) than during the second one (83%).

### 3.2. Characteristics of the Study and Survey Participants

As shown in Table 1, age, sex, and outcome distributions were similar between survey participants and the overall study population. Among survey participants, 42.7% were female, 60.2% were ≥65 years of age, 70.7% had at least one chronic condition, and 36.6% were test-positive cases. A higher proportion of survey participants with at least one chronic condition (83.4%) and of test-positive cases (41.5%) was found among non-participants in the survey.

### 3.3. Precautionary Health Behaviours

#### 3.3.1. Precautionary Health Behaviours Among the Survey Participants

Each survey participant was attributed a PHB composite score from 0 to 7, with a higher score corresponding to a greater level of PHBs. As shown in Figure 1, the median PHB score of survey participants enrolled during the first booster season was higher than for those enrolled during the second booster season. In both seasons, the median PHB score was higher in survey participants ≥65 years old, those with ≥1 chronic condition, those vaccinated in the study period of interest, and test-positive cases. The median score was higher in females in the first booster season, but median scores were similar between sexes in the second booster season. Detailed results of the survey by age group, site, outcome, and exposure are available as Appendix A (Appendix A).

Box and whisker plots map the PHB composite scores, including the minimum and maximum scores (end of whiskers, excluding outliers), the first quartile and third quartile (the bottom and top of the box, respectively), and the median (horizontal line within the box). Outliers are represented by dots. Cases: SARS-CoV-2 test-positive; control: SARS-CoV-2-negative; vaccinated: survey participants having received either a complete primary schedule or at least one booster dose with any COVID-19 vaccine during the study period of interest (first booster or second booster season); unvaccinated in the study period: survey participants having no history of COVID-19 vaccine doses or having received their last dose ≥ 6 months before symptom onset.

In total, 727 survey participants were included in the first booster season analysis (333 test-positive cases and 394 test-negative controls) and 481 in the second booster season analysis (96 test-positive cases and 385 test-negative controls) (Figure 2). Table 2 shows the level of association between each indicator of PHBs and both the exposure (vaccinated with COVID-19) and the outcome (test-positive case) by the study period of interest. In both periods, point estimates for some indicators of PHBs showed 95% CIs excluding unity, suggesting potential associations with being vaccinated in the study period of interest, although these associations were generally not significant when compared to the Bonferroni-adjusted threshold. Several associations were found between indicators of PHBs and being a test-positive case, reaching significant levels, after Bonferroni adjustment, in the first booster season. Overall, no individual PHB statement was significantly associated with both the exposure and the outcome, in either season.

#### 3.3.2. Confounding by PHBs of COVID-19 Vaccine Effectiveness Estimates

COVID-19 VE estimates with and without adjustment for PHBs are presented in Table 3. The median time of the positive-test result since vaccine dose among those vaccinated in the study period of interest was 84 days for the first booster season and 83.5 days for the second booster season. In the first booster season, adjustment for PHBs increased the VE point estimates by six to nine percentage points, depending on the adjustment model used. CIs around the PHB-adjusted point estimates and around the non-adjusted estimates did not overlap. In the second booster season, the increase in VE after adjustment for PHBs was three to four percentage points, with wide and overlapping CIs. For both seasons, adding PHBs to the model significantly improved its goodness-of-fit. The sensitivity analyses using GAMs instead of GEEs as an estimating model or applying other cut-offs when adjusting for PHBs as a binary variable resulted in similar trends, although CIs around point estimates overlapped in this case (Appendix A Appendix A).

## 4. Discussion

In this survey-based research, we show that the level of PHBs among survey participants of a European observational COVID-19 TNCC VE study were modified over time, with lower levels of precautionary attitudes in the second booster season (2022–2023) than in the first COVID-19 booster season (2021–2022). This study was split into three steps: (1) identification of indicators of PHBs as potential confounders, (2) association screening, and (3) model comparison. Significant associations between PHB indicators and outcome were only observed in the first COVID-19 booster season, whereas significant associations between PHB indicators and exposure were rare in both seasons. Although association between a variable and exposure and outcome is required to consider it as a confounder, the absence of statistically significant results does not directly exclude a variable as a potential confounder. A study may fail to demonstrate the association due to limited power (low sample size) or random variation even if the association truly exists. As such, the associations were seen as a screening process, not as a pre-determined criterion. Comparison of PHB-adjusted and non-adjusted VE models suggested confounding of estimates by PHBs in the first booster season, with a six to nine percentage-point increase in VE estimates after including PHBs in the model and no overlap between the CIs obtained around the adjusted and non-adjusted point estimates. Conversely, the results did not suggest PHBs to be a confounding variable in the second booster season.

During the COVID-19 pandemic, various PHBs and other non-pharmaceutical interventions were promoted or made mandatory with the objective to reduce the transmission of SARS-CoV-2 in the population [2,11]. In Europe, adherence to these PHBs differed according to regional and sociodemographic characteristics, with more PHBs among females, older adults, people with poorer health status, and people with a higher perceived severity of COVID-19 [12,13]. People, on average, became less cautious over time but, as in our study, those at the highest risk of severe COVID-19 generally maintained a higher level of PHBs [12]. Although regional and cultural differences are observed with regard to the perception of disease risk and PHBs [14], individual-level factors may have played a greater role in influencing contact behaviour in Europe than country-level factors [15].

In the id.DRIVE study population, PHBs were positively associated with having received a COVID-19 vaccine in the study period of interest, although these were rarely significant after Bonferroni corrections. Associations between PHBs and vaccine exposure have been previously described, with several studies conducted early in the pandemic indicating a higher intent for COVID-19 vaccination in those strongly adhering to PHBs and government guidelines [4,16]. PHBs may also be modified after vaccination, but studies have shown conflicting results. In a bias assessment study analysing national TNCC studies monitoring VE in the United Kingdom, only minimal evidence of riskier behaviour after vaccination was found [17]. Conversely, other studies have shown a decline in adherence to PHBs, an increase in social contacts, or a reduced perception of COVID-19 severity after vaccination [15,18,19]. These observations have been attributed to an increased sense of safety after active immunisation (Peltzman effect) and would lead to a negative association between PHBs and exposure (being vaccinated), the opposite to what we observed in this study.

For the first booster year, we found significant associations between indicators of PHBs (i.e., strongly agree with ‘I avoid touching door handles and staircase railings at public locations’, ‘I would self-isolate myself at home if needed’, and ‘I frequently use hand sanitizer and/or wash my hands after shaking someone’s hand’, and strongly disagree with ‘I do not mind going to very crowded places’) and a SARS-CoV-2-positive result. This result may appear counterintuitive, as adherence to PHBs is expected to reduce viral transmission and the incidence of COVID-19 [5]. However, id.DRIVE focuses on VE against severe disease (hospitalisation for SARI). Older adults, and those with certain predisposing comorbidities or frailty, are at increased risk of severe COVID-19 and hospitalisation [20]. As this population has also been identified as adhering most strongly to PHBs [12,13], this could explain the detected associations between indicators of PHBs and a SARS-CoV-2-positive result. In the second booster year, the general reduction in precautionary behaviour or the decrease in COVID-19 severity due to the emergence of Omicron and later subvariants (including Omicron BA.5, BQ.1, BA.2.75, and XBB; erviss.org) could explain the loss of significance in the associations between PHBs and outcome [21]. The variants circulating during this season and the growing population immunity may also explain the lower VE estimates found for this later period [22]. Indeed, the VE estimates measured in the present study indicate the incremental protection conferred on top of any previous infections or prior vaccinations that were administered 6 months or more before the SARI episode.

Importantly, beside interrupting transmission pathways of SARS-CoV-2, PHBs also interrupt the transmission of other SARI-causing pathogens such as influenza and RSV, which constitute the control group of the TNCC design. If the level of protection conferred by PHBs was similar across all SARI-causing pathogens, no association between PHBs and test-status would be expected. However, interactions are likely complex because the effectiveness of PHBs to mitigate transmission will depend on both the transmissibility and the mode of transmission (direct contact, indirect contact, droplet, and aerosol) specific to each pathogen [23]. The changing epidemiology of these other SARI-causing pathogens, with near-absent circulation during the pandemic-related lockdowns in 2020–2021, off-season waves in 2021–2022, and progressive restoration of pre-pandemic dynamics in 2022–2023 [24,25,26], should be considered in the time-varying changes (significant positive associations after Bonferroni correction during the first booster period and non-significant associations after Bonferroni correction during the second booster period) we observed in the associations between PHBs and outcome.

In our study, adjusting for PHBs led to a six to nine percentage-point increase in VE estimates against COVID-19 hospitalisations in the first booster season. In the second booster season, adjustment for PHBs did not substantially modify VE estimates (three to four percentage-point increase and overlapping CIs) but did continue to improve the model’s goodness-of-fit. In both seasons, no individual PHB statement was found significantly associated with both the exposure and the outcome. This could indicate that only a general comprehensive PHB rather than the adherence to a single preventive measure (e.g., hand-washing) would act as a confounder.

Presumably, it is not only the level of PHB confounding of VE estimates that may differ between seasons (or studies), but also the direction of confounding. Depending on whether positive or negative associations are found between PHBs and exposure or outcome, the level and direction of confounding may change. They may depend on the study outcomes (e.g., SARS-CoV-2 infection and hospitalisation), the setting (e.g., country and cultural specificities towards vaccines and PHBs), the study population (e.g., age, sex, or risk population, vaccination, and outcome probability), the epidemiological context (e.g., endemic, [co-]epidemic, and pandemic settings), and the study design. For example, in a cohort study, exposure (e.g., vaccination) and outcomes (e.g., infection and hospitalisation) are measured over time, which can help control for confounders through design and analysis. By contrast, a TNCC study compares the odds of exposure between those who test positive and those who test negative for the outcome of interest. The dynamics of other respiratory pathogens causing similar symptoms can confound the associations in TNCC designs, as these pathogens might influence both the likelihood of getting tested and the probability of testing positive, thereby impacting the confounding effect of PHBs.

The directed acyclic graph in Figure 3, adapted from Stuurman et al. and Lane et al. [27,28], is a visual representation of the path between the exposure and the outcome, with causal assumptions, covariates, and effect modifiers to consider in addition to PHBs. As previously described, the id.DRIVE COVID-19 VE model adjusts for age, sex, time since vaccination, and number of chronic conditions. It uses a TNCC design, commonly used to assess COVID-19 VE, which reduces unmeasured confounding resulting from general healthcare-seeking behaviour [29]. Pre-existing immunity from prior infection, additional chronic conditions associated with severe COVID-19 disease (e.g., obesity, frailty, and neurological disease) [20], specific health behaviours (e.g., time to health seeking and vaccination against other SARI-causing pathogens), socioeconomic status, and ethnicity were not included in the model due to either insufficient or absence of data. The influence of these unmeasured confounders on our results is unknown and we cannot exclude the possibility that adjusting for PHBs may have corrected unmeasured bias unaccounted for in our model such as frailty.

Other study limitations are to be noted. First, the questionnaire used, although previously applied for assessing PHBs during the COVID-19 pandemic [8], has not been validated, and no standardised tool for the assessment of PHBs is currently available. Second, the self-administered questionnaire was completed at any time during hospital stay, so the survey participants may have been aware of their SARS-CoV-2 test result before completing the survey, which may have influenced their answers through cognitive biases such as outcome or hindsight bias [30]. Third, gathering PHB data adds to the study complexity, and only Italian and Belgian sites participating in id.DRIVE had the local infrastructure and capacity to carry out the PHB survey, leading to a low sample size and results less generalisable to other European countries. Participation rates differed across sites, and overall, the proportion of individuals with at least one chronic condition and those who tested positive was slightly higher among non-participants in the PHB survey. These individuals were excluded from the VE analyses, which could have introduced bias into the results. Finally, id.DRIVE combines retrospective and prospective patient recruitment, but only prospectively enrolled patients could take part in the survey. Therefore, no data were available for the early phase of COVID-19 vaccine roll-out and VE estimates were those of booster vaccines only. Despite these limitations, this study is a real-world example demonstrating that the assessment of and adjustment for PHBs can impact VE estimates that are generated through a TNCC design.

## 5. Conclusions

In conclusion, PHBs may act as potential confounders of VE against COVID-19. In this study, associations were found with both outcome and exposure in the first COVID-19 booster season. Although adjusting for PHBs resulted in a three to nine percentage-point increase in the id.DRIVE VE estimates, the level and direction of confounding may differ in terms of study population, studied outcome, and epidemiological context. PHBs as confounders should be considered when designing a (COVID-19) VE study, and additional research is required to guide researchers on when and how adjustment for PHBs should be applied in the estimating models.

## Figures and Tables

**Figure 1 vaccines-13-01047-f001:**
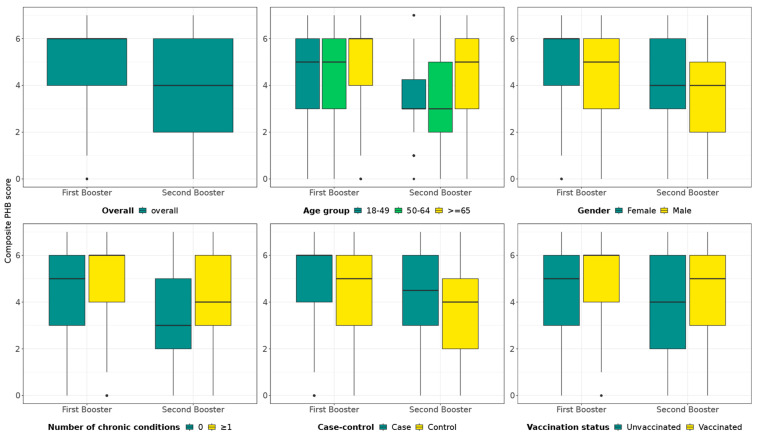
PHB composite score by study period of interest. Data are also stratified by age group, sex, number of chronic conditions, outcome (case or control), and exposure status.

**Figure 2 vaccines-13-01047-f002:**
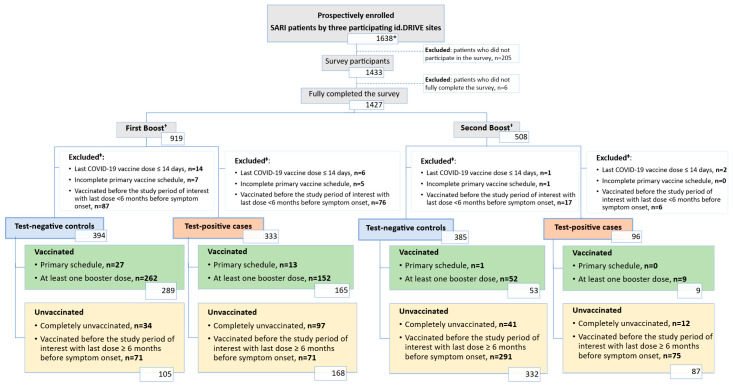
Attrition diagram. Abbreviation: SARI, severe acute respiratory infection. * This cohort was used to describe baseline characteristics (Table 1). ^†^ Stratification based on date of admission. **^‡^** The sum of excluded patients might be greater than the actual number of excluded patients because some patients met multiple exclusion conditions simultaneously.

**Figure 3 vaccines-13-01047-f003:**
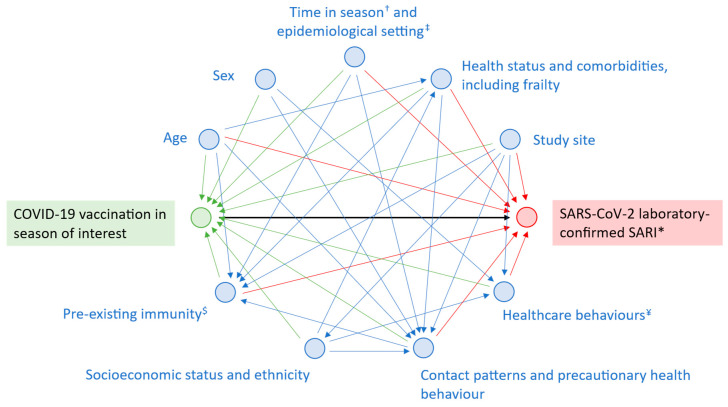
Directed acyclic graph showing the relationship between COVID-19 vaccination and SARS-CoV-2-confirmed SARI. ^†^ Timing of symptom onset or hospitalisation within season (e.g., early autumn versus the end of winter). ^‡^ Endemic, epidemic, or pandemic context; co-circulation of other SARI-causing agents. * Severe acute respiratory infection. ^$^ Infection-acquired or vaccine-induced immunity. ^¥^ Specific healthcare behaviour other than precautionary health behaviour (e.g., time to health seeking, vaccination against other SARI-causing pathogens).

**Table 1 vaccines-13-01047-t001:** Characteristics of the study population.

	Total Study Population N (%)	Survey Participants N (%)	Non-Participants in the Survey N (%)
**Total**	1638	1433	205
**Questionnaire completion**			
Fully completed	1427 (87.1)	1427 (99.6)	0 (0.0)
Incomplete	6 (0.4)	6 (0.4)	0 (0.0)
**Study Contributor**			
CIRI-IT	1332 (81.3)	1262 (88.1)	70 (34.1)
UZA	182 (11.1)	70 (4.9)	112 (54.6)
CHU St Pierre	124 (7.6)	101 (7.0)	23 (11.2)
**Study period of participation**			
First booster season (Nov 2021–Aug 2022)	1022 (62.4)	924 (64.5)	98 (47.8)
Second booster season (Sep 2022–Aug 2023)	616 (37.6)	509 (35.5)	107 (52.2)
**Age group (years)**			
18–49	268 (16.4)	229 (16.0)	39 (19.0)
50–64	393 (24.0)	341 (23.8)	52 (25.4)
≥65	977 (59.6)	863 (60.2)	114 (55.6)
**Number of chronic conditions ***			
None	438 (26.7)	405 (28.3)	33 (16.1)
≥ 1 chronic condition	1184 (72.3)	1013 (70.7)	171 (83.4)
No information	16 (1.0)	15 (1.0)	1 (0.5)
**Sex**			
Male	939 (57.3)	819 (57.2)	120 (58.5)
Female	696 (42.5)	612 (42.7)	84 (41.0)
Missing	3 (0.2)	2 (0.1)	1 (0.5)
**Outcome**			
Test-positive	609 (37.2)	524 (36.6)	85 (41.5)
Test-negative	1029 (62.8)	909 (63.4)	120 (58.5)

Abbreviations: CHU, Centre Hospitalier Universitaire; CIRI-IT, Centro Interuniversitario per la Ricerca sull’Influenza e le Altre Infezioni Trasmissibili; N, number; UZA, Antwerp University Hospital. * Among asthma, lung disease, cardiovascular disease, hypertension, chronic kidney disease, chronic liver disease, type 2 diabetes, cancer, and immunodeficiency.

**Table 2 vaccines-13-01047-t002:** Association between PHB indicators and both exposure and outcome, overall and stratified by time periods of interest.

PHB Indicator ^†^	Association Between PHB Indicator and Vaccination in Study Period of Interest		Association Between PHB Indicator and SARS-CoV-2 Test-Positive Outcome	
	OR (95% CI) ^‡^	*p*-Value	OR (95% CI) ^$^	*p*-Value
**First booster period**				
Strongly agree with ‘It really bothers me when people sneeze without covering their mouth’	2.54 (1.56; 4.15) **	0.0001	1.38 (0.85; 2.27)	0.2025
Strongly agree with ‘I avoid touching door handles and staircase railings at public locations’	1.15 (0.83; 1.61)	0.4135	2.02 (1.45; 2.82) **	<0.0001
Strongly disagree with ‘I do not mind going to very crowded places’	1.55 (1.09; 2.21) *	0.0137	1.8 (1.26; 2.60) **	0.0008
Strongly agree with ‘I would self-isolate myself at home if needed’	1.48 (1.00; 2.2) *	0.0403	1.95 (1.30; 2.95) **	0.0008
Strongly agree with ‘I frequently use hand sanitizer and/or wash my hands after shaking someone’s hand’	1.65 (1.14; 2.41) *	0.0067	2.05 (1.39; 3.06) **	0.0001
Strongly agree with ‘I avoid going to public places’	1.39 (1.01; 1.91) *	0.0396	1.51 (1.10; 2.07) *	0.0089
Strongly disagree with ‘I dislike wearing a face mask because of the way it looks and/or feels’	0.66 (0.46; 0.93) *	0.0173	1.25 (0.88; 1.78)	0.1995
**Second booster period**				
Strongly agree with ‘It really bothers me when people sneeze without covering their mouth’	1.97 (0.81; 5.80)	0.1437	1.08 (0.57; 2.14)	0.8787
Strongly agree with ‘I avoid touching door handles and staircase railings at public locations’	1.26 (0.71; 2.23)	0.4155	1.91 (1.18; 3.09) *	0.0059
Strongly disagree with ‘I do not mind going to very crowded places’	2.54 (1.41; 4.69) **	0.0010	1.64 (1.02; 2.65) *	0.0391
Strongly agree with ‘I would self-isolate myself at home if needed’	1.47 (0.66; 3.73)	0.3739	2.01 (0.98; 4.56)	0.0501
Strongly agree with ‘I frequently use hand sanitizer and/or wash my hands after shaking someone’s hand’	1.13 (0.63; 2.03)	0.6830	1.85 (1.13; 3.09) *	0.0112
Strongly agree with ‘I avoid going to public places’	2.05 (1.16; 3.67) *	0.0087	1.85 (1.15; 2.99) *	0.0077
Strongly disagree with ‘I dislike wearing a face mask because of the way it looks and/or feels	1.25 (0.69; 2.25)	0.4685	0.59 (0.34; 1.01)	0.0512

Abbreviations: CI, confidence interval; OR, odds ratio; PHB, precautionary health behaviour; SARS-CoV-2, severe acute respiratory syndrome coronavirus 2. ^†^ The reference category for the PHB indicator varies depending on the direction of the questions. For questions 1, 2, 4, 5, and 6, ‘strongly agree’ is compared to ‘strongly disagree’ or ‘undecided’ as the reference category. For questions 3 and 7, ‘strongly disagree’ is compared to ‘strongly agree’ or ‘undecided’. ^‡^ Reference category: unvaccinated in the study period of interest. ^$^ Reference category: SARS-CoV-2 test-negative outcome; Significant *p*-value: * without Bonferroni correction (*p*-value < 0.05); ** after Bonferroni correction (*p*-value < 0.0036).

**Table 3 vaccines-13-01047-t003:** COVID-19 VE estimates with and without adjustment for PHBs.

	COVID-19 VE, GEE Reference Model ^a^ % (95% CI)	COVID-19 VE, GEE PHB-Adjusted Model 1 ^b^ % (95% CI)	Percentage Change Between GEE Reference Model ^a^ and PHB-Adjusted Model 1 ^b^	*p*-Value *	COVID-19 VE, GEE PHB-Adjusted Model 2 ^c^ % (95% CI)	Percentage Change Between GEE Reference Model ^a^ and PHB-Adjusted Model 2 ^c^	*p*-Value *
First booster season	46.4 (43.0; 49.6)	55.1 (50.4; 59.4)	18.75%	<0.001	52.6 (47.0; 57.5)	13.36%	<0.001
Second booster season	32.0 (15.9; 45.1)	36.0 (20.2; 48.6)	12.50%	<0.001	35.1 (18.5; 48.3)	9.69%	<0.001

Abbreviations: CI, confidence interval; COVID-19, coronavirus disease 2019; GEE, generalised estimating equation; OR, odds ratio; PHB, precautionary health behaviour; VE, vaccine effectiveness; ^a^ GEE reference model: GEE-based model, adjusted for symptom-onset date, sex, age, and number of chronic conditions; ^b^ GEE precautionary health behaviour-adjusted model 1: GEE reference model with the PHB composite score fitted as a continuous variable (the OR of the PHB composite score in this model is 1.23 (95%CI: 1.19; 1.28) for the first booster season and 1.14 (95% CI: 1.13; 1.16) for the second booster season); ^c^ GEE precautionary health behaviour-adjusted model 2: GEE reference model with the precautionary health behaviour composite score fitted as a binary variable using a cut-off of 5 (the OR of the PHB composite score in this model is 2.57 (95% CI: 1.88; 3.51) for the first booster season and 1.59 (95% CI: 1.49; 1.70) for the second booster season). * Likelihood ratio test *p*-value. *p*-values < 0.05 indicate that adding PHBs significantly improves the model fit.

## Data Availability

De-identified data that underlie the results reported in this article (text, tables, figures, and appendices) may be obtained in accordance with the id.DRIVE data access policy.

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
