# Peer review of "Precautionary Health Behaviours as Potential Confounders in COVID-19 Vaccine Effectiveness Studies"

_vaccines, 2025, doi:10.3390/vaccines13101047_

Round 1
Reviewer 1 Report
Comments and Suggestions for Authors
Precautionary health behaviour as a potential confounder in COVID-19 clinical effectiveness studies clinical trials
- Positive association between PHBs and vaccination (exposure)
General comments:
Decide if the abbreviation for ‘precautionary health behaviours’ is PHB or PHBs – then be consistent in use
Use of ‘subject’ – people or participants
Abstract – Method not quite right regarding use of Fisher’s exact test – need to state without and with Bonferroni adjustment
This needs to be clear throughout document, as in sentence starting line 245: In both periods, indicators of PHB were positively associated with being vaccinated in the study period of interest, although these associations were generally not significant when applying Bonferroni correction.
Results: “PHBs evolved over time” – evolved implies a positive change (that develops gradually) so replace with another word eg were modified, reduced
- in 2nd line of Discussion as well
First booster season: PHBs were POSITIVELY associated with exposures and outcomes – definitions ie vaccination status; COVID-19 infection
Introduction
Good
2nd Para, 3rd line: spectrum of [both] government-initiated, medically- or self-initiated
Lines 85 to 93 - delete
Materials and Methods
Study Population and setting, 3rd last line: European Centre for Disease Prevention and Control (ECDC)
Line 116: Persons/People?
Precautionary Health Behaviour Survey: ”Patients could participate to the survey” – replace with eg Patients could undertake/complete the survey
Line 130: scoreS
Exposure definitions
Line 138: use of ‘subject’ – person? Similarly for other places where the term is used
Statistical methods
Thorough
Controls – non-SURVEY participants? At the 3 sites – state clearly who the controls are
Results
Figure 1: case control not mentioned in Legend
Para beginning line 232 – part of legend?
Line 245 states: In both periods, indicators of PHB were positively associated with being vaccinated (EXPOSED) in the study period of interest, although these associations were generally not significant when applying Bonferroni correction.
Several associations were found between indicators of PHB and being a test-positive case, reaching significant levels in the first booster season – ANY CORRECTIONS made?
Figure 2, Table 2: good
3.3.2. Confounding of PHB on COVID-19 vaccine effectiveness estimates
Line 269: Median time OF THE POSITIVE-TEST RESULT since VACCINE dose among those vaccinated in the study period of interest was 84 days for the first booster season and 83.5 days for the second STUDY PERIOD/one
Line 275: CIs
Discussion
Line 290: PHBs or PHB?
Line 293: Significant associations between PHB indicators and outcome were only observed in the first COVID-19 booster season, whereas significant associations between PHB indicators and exposure were rare in both seasons. – consistency with Results section?
- Authors need to be clear about when they are applying Bonferroni correction and when they are not. I suggest providing findings without and with correction – as in para beginning Line 311
Line 321: These observations have been attributed to an increased sense of safety after active immunisation (Peltzman effect) and would lead to a negative association between PHB and exposure, - TO COVID-19? IE DIFFERENT USE OF ‘EXPOSURE’ TO HOW DEFINED IN THIS STUDY (vaccinated)
Para beginning Line 325: important
Line 330: As this population has also been identified as adhering most strongly to PHB [11, 12], this could explain the detected associations BRIEF EXPLANATION OF WHAT THESE ARE?
Sentence beginning Line 337: Indeed, the VE estimates measured in this /THE PRESENT? study indicate the incremental protection conferred on top of any previous infections or prior vaccinations that were administered 6 months or more before the SARI episode.
Line 340: PHB also interrupt(s) the transmission
Line 350: the time-varying changes we observed in the associations between PHB and outcome – useful to state what these were and their levels of significance (with and without applying Bonferroni correction
Line 387: we cannot exclude THE POSSIBILITY that adjusting for PHB may have corrected unmeasured bias unaccounted for in our model
Limitations – well identified
Conclusions
Line 418: Although adjusting for PHB resulted in a minor increase of the id.DRIVE VE estimates, - BETTER TO STATE WHAT THE DIFFERENCE WAS RATHER THAN CALLING IT ‘MINOR’?
Author Response
Please see an attachment.

Reviewer 2 Report
Comments and Suggestions for Authors
Dear authors,
The current manuscript aimed to investigate the potential confounding of
Precautionary health behaviours (PHBs), such as hand-washing or self-isolation, on COVID-19 vaccine effectiveness (VE) estimates in a subset of study participants enrolled in id.DRIVE (European multicentre test-negative case–control study estimating COVID-19 VE). VE estimates were generated with and without adjustment for PHB. Results clarified that PHBs evolved, with higher precautionary attitudes in the first COVID-19 vaccine booster season (2021–2022) compared to the second one (2022–2023). Adjusting for PHB resulted in a 6-9% increase in VE estimates for the first dose, and no confounding by PHB was observed in the second booster season. The authors concluded that PHBs should be considered a possible confounder of COVID-19 VE studies. Hand-washing and self-isolation are protective measures against any infection, not a confounding factor for vaccine efficacy. They protect against infection or deterioration of cases but do not increase or decrease the efficacy of any vaccine. The authors clarified that in the background.
I have some comments:
- The background includes a paragraph that seems to be from ChatGPT (from line 85 to 93).
- The study design just indicates the source, not the design.
- The study population and setting are not clear, and also, how many from each place and their inclusion criteria.
- Most ethical approvals are mentioned without dates in the methods section.
- Authors mentioned that Informed consent was provided by all participants or their legally acceptable representative(s). While they said the age group is >18 years, why is there consent from the legally acceptable representatives?
- Many of the results mentioned lack the P value.
- Other comments in the attached manuscript.
Best regards,

Author Response
Please see an attachment.

Reviewer 3 Report
Comments and Suggestions for Authors
Dear Authors, Dear Editor,
Thank You for the ability to review the manuscript Precautionary health behaviour as a potential confounder in COVID-19 vaccine effectiveness studies.
The authors decided to elucidate whether PHBs should be considered as a potential confounder. This is an important issue for all type vaccine effectiveness studies. The topic is challenging and requires methodological strictness which is not visible to be achieved in the submitted manuscript.
In my opinion the following issues should be considered to improve the scientific soundness, the concept and readability of the manuscript:
- In the manuscript PHBs is considered as a single variable. It should be better justified the construct of PHBs, why only 7 items, why just the items presented in the manuscript?
- The confounding effect is a specific type of variable interplay, which must meet all 3 criteria: 1) the variable must be statistically associated with the exposure; 2) the variable must cause the outcome, and 3) the variable must not be on a causal pathway => the description presented in the manuscript lines 70-71 is not correct and may provide confusions. I suggest correcting this.
- Lines 85-93 is a copy of manuscript preparation guidelines … in my opinion.
- As the title clearly states PHB as a potential confounder – describe clearly under study purposes the steps planned to be done to verify that hypothesis. Include, please, the criteria for a confounder.
- To my mind the analytical criteria for PHB as a confounder are not clearly presented. Criterion 1 is not visible, Table 2 presents some statistical results, but only a few, and the models are not adjusted for other important covariates used in table 3.
- Table 3 presents estimates of the whole models. The model/s fit has/have been improved, but it is not visible what is an impact of PHB itself. Thus it is not clearly visible the second criterion.
- I suggest adding also analyses showing the PHBs are not on the causal pathway.
- Additionally, I suggest showing stratified analyses to assure, at least partially, that there is no other / not measured or not considered / variable which produced percentage change effect presented in table 3.
- As the ability to conclude whether the PHBs has been identified as a confounder depends on the mentioned above analyses I do not comment the quality of general conclusions.
Reviewer
Comments on the Quality of English LanguageTo my understanding a confounder is a variable that creates a spurious association / or that distorts the true relationship; a confounding effect is a spurious/distorting effect; confounding of something is the process which distorts this ‘something’, consequently confounding of PHB would be a process which distorts the PHB – this was not, however, the concept presented by the authors. In my opinion the use of sentence ‘confounding of PHBs’ is confusing and may lead to misunderstanding. I suggest correcting this across the whole manuscript.
Round 2
Reviewer 2 Report
Comments and Suggestions for Authors
Responses are satisfactory
Author Response
Dear Reviewer 2,
Thank you for your second review round. We are glad to know our responses were satisfactory and that there are no further comments.
The authors
Reviewer 3 Report
Comments and Suggestions for Authors
Dear Authors,
The submitted revised version of the manuscript addressed almost all of my comments and issues requiring clarification. I have still, however, a concern about the presented criteria required for a variable to be considered a confounder … I described/cited in my first review, in the 2nd point: “the variable must cause the outcome”. In the current version of the manuscript you put “must be associated”. The association, however, is not the same as causal relationship. I understand the data from the study does not justify statements on causality, and I’d like to express my endorsement for the final conclusion stating “PHB may act as a potential confounder ...”, however, the statement presented under Statistical methods (lines 155-158) imitates theoretical considerations and in my opinion should be consistent with a theory in that matter. In summary, I strongly suggest correcting sentence 2) line 156.
Reviewer
Author Response
Dear Reviewer 3,
Thank you very much for your comment, and all your previous feedback, it improves our manuscript greatly. Please find attached the new version with that sentence corrected.
The authors
